# Initial Implementation and Utilization of Cardiopulmonary Exercise Testing at a Pulmonary Department of an Academic Tertiary Care Center: An Overview

**DOI:** 10.3390/jcm14113676

**Published:** 2025-05-23

**Authors:** Nimrod Kleinhaus, Yael Raviv, Itamar Ben Shitrit, Jonathan Wiesen, Liora Boehm Cohen, Michael Kassirer, Natalya Bilenko

**Affiliations:** 1Joyce and Irving Goldman Medical School, Faculty of Health Sciences, Ben-Gurion University of the Negev, Beer Sheva 8410501, Israel; 2Department of Pulmonology, Soroka University Medical Center, Faculty of Health Sciences, Ben-Gurion University of the Negev, Beer Sheva 8410501, Israel; jwiesen1@gmail.com (J.W.); lioraboehmcohen@gmail.com (L.B.C.); michaelkas@clalit.org.il (M.K.); 3Department of Epidemiology, Biostatistics and Community Health Sciences, School of Public Health, Faculty of Health Sciences, Ben-Gurion University of the Negev, Beer-Sheva 8410501, Israel; natalya@bgu.ac.il

**Keywords:** cardiopulmonary exercise testing, anerobic threshold, respiratory exchange ratio

## Abstract

**Background:** Cardiopulmonary exercise testing (CPET) is a valuable diagnostic and prognostic tool for assessing the integrated function of the cardiopulmonary and muscular systems during exercise. The initiation of a CPET program is complex, and data on early implementation in academic centers remain relatively limited. **Objective:** to evaluate the initial integration of CPET within a pulmonary department, focusing on patient demographics, referral indications, test performance, and factors associated with anaerobic threshold achievement. **Methods:** A retrospective cohort study was conducted at a single tertiary care center, including all patients who underwent their first CPET between February 2016 and December 2022. Demographic, clinical, and functional parameters were extracted. Multivariable logistic regression was used to identify variables associated with anaerobic threshold achievement, defined as a respiratory exchange ratio (RER) ≥ 1.1. **Results:** The cohort included 434 patients (mean age 60.3 ± 14.1 years; 54% male; mean BMI 29.2 ± 5.6 kg/m^2^). The most common indication for testing was dyspnea (50%). Tests were most frequently terminated due to leg discomfort (39%) and dyspnea (38.8%). Achievement of RER ≥ 1.1 was independently associated with lower BMI (aOR = 0.91; 95% CI: 0.88–0.95; *p* < 0.001), higher FVC % predicted (aOR = 1.02; 95% CI: 1.00–1.03; *p* = 0.028), and greater minute ventilation volume (aOR = 1.02; 95% CI: 1.01–1.03; *p* < 0.001), and it was less likely in patients referred for cardiovascular disease (aOR = 0.37; 95% CI: 0.21–0.64; *p* < 0.001). No consistent temporal trend in RER achievement was observed across the study period. **Conclusions:** CPET was most commonly utilized in response to patient-reported dyspnea, with test termination frequently driven by subjective symptoms rather than objective clinical criteria. Anaerobic threshold achievement was more strongly associated with individual physiological characteristics than with institutional experience. These findings underscore the importance of patient preparation and pulmonary functional capacity in optimizing CPET performance.

## 1. Introduction

Cardiopulmonary exercise testing (CPET) is a comprehensive diagnostic modality that surpasses conventional exercise testing by incorporating measurements of ventilation, oxygen uptake (VO_2_), and carbon dioxide output (VCO_2_). This approach provides an integrated assessment of physiological responses to exercise, capturing the interplay among cardiac, pulmonary, and musculoskeletal systems. CPET serves diverse clinical purposes, including the evaluation of unexplained exertional dyspnea, quantification of exercise capacity, and preoperative risk stratification in some patients. It is particularly valuable in the management of patients with established cardiopulmonary conditions such as chronic obstructive pulmonary disease (COPD), pulmonary vascular disease, and congestive heart failure (CHF). While CPET does not typically yield a definitive diagnosis, it refines the differential diagnosis—especially when resting evaluations are inconclusive—and can reduce the need for further diagnostic procedures when findings are within normal limits. Moreover, CPET enables clinicians to assess disease severity, monitor treatment efficacy, and tailor rehabilitation or exercise interventions to individual patient needs. In individuals with multiple comorbidities, CPET facilitates the identification of the primary source of functional limitation and supports informed therapeutic planning [1,2,3,4,5,6].

Prior to undergoing CPET, patients are instructed to abstain from vigorous physical activity, heavy meals, caffeine, and smoking on the day of the test. During preparation, the patient is dressed in appropriate exercise attire and equipped with a mask or mouthpiece for respiratory gas analysis, electrocardiogram (ECG) leads, a blood pressure cuff, and a pulse oximeter for continuous monitoring of physiological parameters. The test is conducted on a treadmill or cycle ergometer, beginning with a low-intensity warm-up and progressively increasing in difficulty until the patient reaches volitional exhaustion, typically within 8 to 12 min. This is followed by a brief cool-down and recovery phase. The data collected are subsequently analyzed to evaluate the patient’s cardiopulmonary performance and physiological responses under exertion [3,7,8].

As with other respiratory assessments, the accuracy and diagnostic utility of CPET are highly dependent on the patient’s level of effort and cooperation. Achieving maximal exertion is essential to ensure the validity and interpretability of the test results. Key physiological markers, such as the anaerobic threshold (AT), can only be reliably identified when the patient surpasses this threshold of metabolic transition. The AT represents a critical parameter in the evaluation of physical fitness in healthy individuals and serves as an important indicator of potential cardiovascular limitations [1,9,10].

Several methods, both invasive and non-invasive, are available for determining the AT [1]. One widely accepted non-invasive approach involves the calculation of the respiratory exchange ratio (RER), defined as the ratio of VCO_2_ to VO_2_. An RER value of ≥1.1 is conventionally used to indicate the onset of anaerobic metabolism and is considered a reliable marker of maximal patient effort. This threshold has been validated across numerous studies and is broadly recognized in the CPET literature for its clinical relevance in confirming adequate exertion [2,9,11,12,13]. RER is typically derived from breath-by-breath gas exchange measurements collected during exercise [7].

The aim of this study is to characterize and evaluate the initial implementation of CPET within the pulmonary department. Specifically, we aim to examine how CPET has been integrated into routine clinical practice and to analyze patterns of patient utilization. Variable selection was designed to capture both clinical and patient-centered dimensions of CPET use, thereby providing a comprehensive overview of its role in real-world settings. As a secondary objective, we investigate the association between patient and test characteristics and the achievement of an RER ≥ 1.1, in order to optimize CPET utilization in clinical practice. We hypothesize that rates of AT achievement will increase over time as CPET becomes more established in clinical practice.

## 2. Methods

This single-center, retrospective cohort study was conducted at the Pulmonary Department of Soroka University Medical Center (SUMC) in Beer-Sheva, Israel. Data were manually extracted from electronic medical records covering the period from February 2016 to December 2022. This study was approved by the institutional Helsinki Committee of SUMC (protocol code: 0366-21-SOR) and adheres to the Strengthening the Reporting of Observational Studies in Epidemiology (STROBE) guidelines [14].

The analysis included adult patients (≥18 years) who underwent their first CPET at the Pulmonary Department of SUMC between February 2016 and December 2022. Inclusion was limited to patients whose CPET data were complete and interpretable according to standard reporting criteria. Exclusion criteria included repeated CPETs, tests terminated prematurely due to technical issues, and tests with missing or invalid measurements.

The primary outcome—characterizing CPET utilization patterns—was evaluated based on patient demographics, referral indications, rates of RER achievement, and reasons for test termination.

The secondary outcome was the association between patient and test characteristics and the achievement of the anaerobic threshold, defined as an RER ≥ 1.1.

Independent variables included demographic, clinical, pulmonary function, and CPET-specific parameters. Demographic variables encompassed the year of CPET performance, sex, age, and body mass index (BMI). The clinical indication for testing was also recorded, reflecting the primary reason for referral.

Pulmonary function variables included forced vital capacity (FVC, % predicted), forced expiratory volume in one second (FEV_1_, % predicted), and the FEV_1_/FVC ratio, which reflects the proportion of volume exhaled in the first second relative to total exhaled volume. Additional parameters included total lung capacity (TLC, % predicted), functional residual capacity (FRC, % predicted), residual volume to total lung capacity ratio (RV/TLC, % predicted), diffusing capacity for carbon monoxide (DLCO, % predicted), and maximal voluntary ventilation (MVV, mL/min).

CPET-derived variables included oxygen uptake (VO_2_, mL/kg/min), ventilation rate (L/min), respiratory rate (breaths/min), tidal volume (mL), and the ventilation-to-carbon dioxide output ratio (VE/VCO_2_). Additional measures recorded were end-tidal carbon dioxide pressure (PetCO_2_, mmHg), oxygen saturation (%), blood pressure (mmHg), heart rate (beats/min), and oxygen pulse (mL/beat), defined as oxygen uptake per heartbeat. The reason for test termination was also documented, distinguishing between physiological or clinical indications and patient-reported limitations.

Data were obtained from CPET interpretation reports, the CPET system database, pulmonary function test results, and individual patient electronic medical records. All data were manually reviewed and extracted by the research team.

The study size was determined by the total number of eligible patients who completed CPET at the pulmonary department during the study period. As all qualifying cases were included without sampling, the sample size reflects the full available cohort, enhancing representativeness and minimizing selection bias.

## 3. Statistical Methods

In descriptive statistics, continuous variables were presented as means and standard deviations (SDs). Categorical variables were summarized as counts and percentages. We used univariate analysis, and we employed Pearson’s chi-squared test, the Wilcoxon rank sum test, and Fisher’s exact test. To determine demographic and clinical association for reaching RER ≥ 1.1, a multivariate logistic regression analysis was performed. Variables were included in the model if they demonstrated a significance level of *p* < 0.05 in univariate analysis or were considered clinically relevant. Collinearity assessments were conducted to ensure predictor independence. To ascertain the most robust model, comparisons were made using the Bayesian information criterion (BIC), alongside the application of backward elimination techniques. All statistical analyses were performed with a significance level of α = 0.05 (two-sided) using R Studio version 4.3.2.

## 4. Results

A total of 474 patients underwent their first cardiopulmonary exercise test (CPET) during the study period. Following application of the exclusion criteria, 434 tests remained eligible for analysis (Figure 1). The cohort had a mean age of 60.3 years (SD 14.1) and a mean body mass index (BMI) of 29.2 kg/m^2^ (SD 5.6), with males comprising 54% of the population (Table 1).

The most common referral indication was unexplained shortness of breath (dyspnea), reported in 50% of cases (*n* = 415), followed by respiratory diseases (*n* = 224, 27%) and cardiovascular conditions (*n* = 104, 12.5%) (Figure 2). Notably, dyspnea was cited more frequently than all respiratory disease diagnoses combined. Patients could have more than one indication for testing. Among those referred due to respiratory diseases, the most frequently documented conditions were current or past smoking (*n* = 78) and pulmonary hypertension (*n* = 45).

The primary reasons for test termination (Figure 3) were leg discomfort (*n* = 232, 39%) and shortness of breath (dyspnea) (*n* = 231, 38.8%). Only a small proportion of participants (*n* = 16, 2.7%) discontinued the test upon reaching maximal exertion, as defined by clinical criteria. Multiple reasons for termination could be recorded for a single test.

In a multivariable logistic regression analysis (Table 2), achievement of the AT—defined as a respiratory exchange ratio (RER) ≥ 1.1—was significantly associated with several variables. These included lower BMI (adjusted odds ratio [aOR] 0.91, 95% CI: 0.88–0.95, *p* < 0.001), referral for cardiovascular indications (aOR 0.37, 95% CI: 0.21–0.64, *p* < 0.001), higher forced vital capacity (FVC, % predicted) (aOR 1.02, 95% CI: 1.00–1.03, *p* = 0.028), and greater minute ventilation volume (MVV) (aOR 1.02, 95% CI: 1.01–1.03, *p* < 0.001). The year of the test was not significantly associated with RER achievement and was excluded from the final model during the model selection process.

Figure 4 illustrates the relationship between test year and AT achievement (RER ≥ 1.1) during CPET between 2016 and 2022. Contrary to the study hypothesis, no upward trend was observed over time. The highest achievement rate occurred in 2017 (51.0%), followed by a marked decline in 2018 (20.3%). Rates subsequently increased in 2019 (38.6%) and remained relatively stable in 2020 (46.0%) and 2021 (46.8%), before decreasing again in 2022 (33.3%). While year-to-year variability is evident, no consistent directional trend emerged.

Appendix A presents the number of CPETs performed annually, alongside the distribution of variables identified as significant in the logistic regression model (Table 2). However, year-by-year comparisons of these variables do not fully explain the observed fluctuations in anaerobic threshold achievement rates over time.

## 5. Discussion

The study population was relatively evenly distributed between male and female participants, with a mean age and BMI higher than that of the general Israeli population [15]. This demographic profile is consistent with prior research demonstrating a high prevalence of overweight and obesity among older adults, particularly those with chronic respiratory conditions [16,17,18]. These findings suggest that our cohort is representative of the typical patient population referred to pulmonary departments for CPET evaluation. Moreover, the demographic characteristics observed in our study are in line with those reported in other related studies, reinforcing the generalizability of our results [19,20,21].

Shortness of breath was the most common indication for CPET referral in our cohort, followed by respiratory and cardiovascular diseases. This distribution underscores the primary utility of CPET as a diagnostic tool frequently initiated in response to subjective symptoms. These findings are consistent with existing literature that highlights the role of CPET in the evaluation of unexplained dyspnea, particularly in cases where standard diagnostic assessments fail to identify a clear etiology [21,22].

Our findings indicate that CPET is most commonly terminated due to subjective symptoms—such as leg discomfort, dyspnea, and fatigue—rather than objective clinical criteria like abnormal ECG changes or hypertensive responses. This observation is consistent with existing evidence suggesting that non-physiological factors often limit exercise performance during CPET [7]. It highlights the influence of psychological variables, including anxiety, motivation, and lack of test familiarity, on patient outcomes—findings that are well documented in the literature [23,24]. Interventions aimed at addressing these barriers, such as structured pre-test counseling or patient education, may improve both performance and the diagnostic accuracy of CPET [25].

We observed no consistent trend in anaerobic threshold achievement rates across the study period, suggesting that factors beyond cumulative institutional experience play a substantial role in determining test outcomes. While an initial increase in RER ≥ 1.1 achievement was noted between 2016 and 2017—potentially reflecting improved staff proficiency—this was followed by a marked decline in 2018, which could not be fully explained by patient characteristics. This pattern points to the possible influence of unmeasured external or procedural variables. Future qualitative studies examining aspects such as patient preparation, technician training, and protocol adherence may help clarify these fluctuations.

Notably, during the COVID-19 pandemic years (2020–2021), achievement rates remained relatively high (approximately 46%) [26]. This may reflect selection bias, as individuals with greater health awareness or fewer comorbidities were more likely to present for testing, consistent with trends reported in other areas of clinical care during the pandemic [27]. These findings raise the possibility that patient performance during CPET may be influenced by a combination of psychological, procedural, and fixed individual factors, rather than institutional experience alone.

The multivariable analysis identified significant associations between achieving a RER ≥ 1.1 and several physiological parameters, including lower BMI, higher predicted FVC, and greater MVV. These findings highlight the importance of pulmonary capacity and overall physical fitness in influencing CPET performance [21,28,29,30,31,32,33]. Logistic regression analysis incorporating demographic, clinical, and pulmonary function variables enabled the identification of patient-level predictors of test success. This approach offers clinicians valuable insights for tailoring patient preparation and optimizing test conditions. In contrast, a referral indication of cardiovascular disease was independently associated with reduced likelihood of reaching the anaerobic threshold, likely reflecting exercise limitations imposed by underlying cardiac dysfunction [2,32].

A key limitation of this study is the lack of data on technician-related variability, which may introduce confounding bias. However, the potential impact of this limitation was mitigated by the fact that all CPETs were conducted by the same two technicians throughout the study period, promoting procedural consistency. Additionally, while the analysis focused on RER as the primary marker of maximal exertion, future research should consider incorporating complementary measures such as VO_2_ max, patient-reported experience, and diagnostic yield to provide a more comprehensive assessment of CPET utility.

The strengths of this study include its comprehensive inclusion of all eligible patients undergoing CPET at the pulmonary department, thereby reducing selection bias. Furthermore, the diverse patient population at SUMC enhances the external validity of the findings, supporting their applicability to a wide range of clinical settings.

## 6. Conclusions

By analyzing patient demographics, referral indications, and test performance, this study offers a comprehensive overview of the initial integration of CPET into clinical practice within a pulmonary department of a tertiary care center. The findings confirm CPET’s primary role as a diagnostic tool prompted largely by patient-reported symptoms, rather than objective clinical findings. The frequent termination of tests due to subjective factors—such as leg discomfort and dyspnea—highlights the combined influence of physiological and psychological limitations on test outcomes.

Notably, achievement of the AT (defined as RER ≥ 1.1) did not demonstrate a consistent improvement over time, suggesting that institutional experience alone does not drive enhanced performance. Instead, RER achievement was associated with lower BMI, higher predicted FVC, and greater MVV, reflecting the impact of individual pulmonary and functional capacity. In contrast, the presence of cardiovascular disease was negatively associated with test completion at anaerobic threshold levels. These findings support the clinical utility of CPET in pulmonary evaluation and underscore the need for targeted strategies—such as patient preparation and procedural consistency—to optimize test accuracy and interpretability.

## Figures and Tables

**Figure 1 jcm-14-03676-f001:**
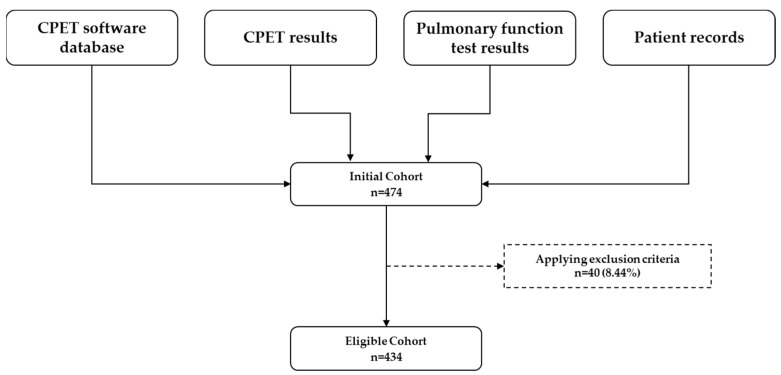
Data sources and selection process for the eligible cohort. CPET = cardiopulmonary exercise test.

**Figure 2 jcm-14-03676-f002:**
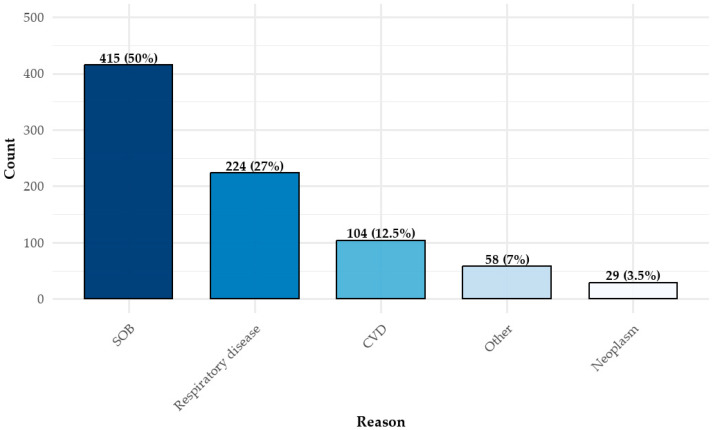
Reasons for CPET administration *. * There can be more than one reason. CPET = cardiopulmonary exercise test; SOB = shortness of breath (dyspnea); CVD = cardiovascular disease.

**Figure 3 jcm-14-03676-f003:**
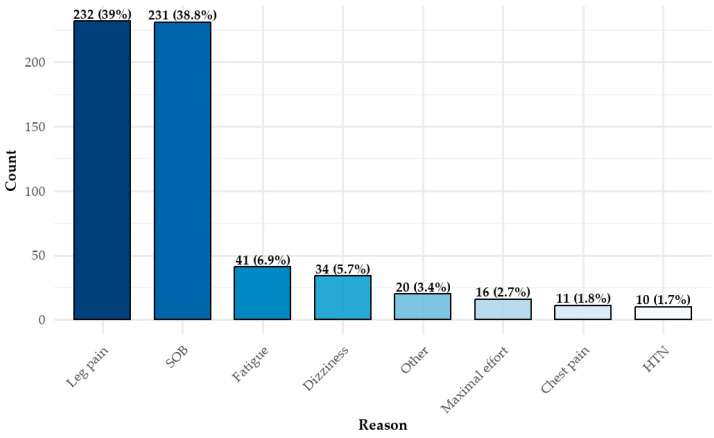
Distribution of CPET termination reasons *. * There can be more than one reason. CPET = cardiopulmonary exercise test; SOB = shortness of breath (dyspnea); HTN = hypertension.

**Figure 4 jcm-14-03676-f004:**
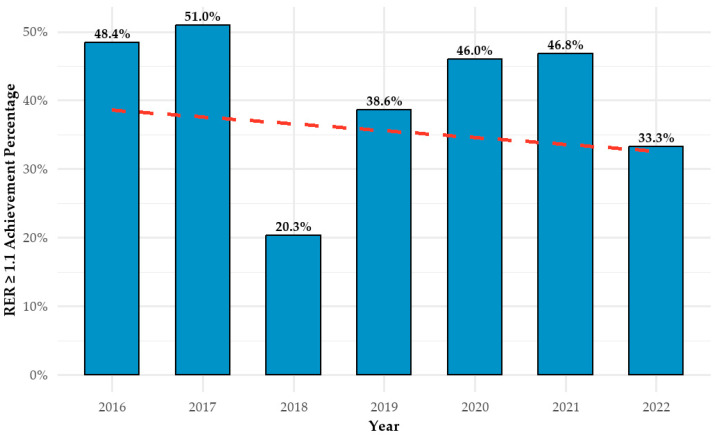
Annual CPET anaerobic threshold achievement rates with trendline. CPET = cardiopulmonary exercise test; AT = anaerobic threshold.

**Table 1 jcm-14-03676-t001:** Demographic and Clinical Characteristics of the CPET Patient Population.

Characteristic	N = 434
Sex (Male) *n*/N (%)	231/425 (54%)
Age, Mean (SD)	60.27 (14.05)
BMI (kg/m^2^), Mean (SD)	29.19 (5.63)

BMI = body mass index; N = number; SD = standard deviation.

**Table 2 jcm-14-03676-t002:** Association of RER ≥ 1.1 Achievement—A Multivariate Logistic Regression Study (N = 423).

Variable	aOR	95% CI	*p*-Value
BMI (continuous)	0.91	0.88, 0.95	<0.001
Cardiovascular Disease (dichotomous)	0.37	0.21, 0.64	<0.001
Forced Vital Capacity (continuous)	1.02	1.00, 1.03	0.028
Minute Ventilation Volume (continuous)	1.02	1.01, 1.03	<0.001

BMI = body mass index; aOR = adjusted odds ratio; CI = confidence interval.

## Data Availability

The main document contain the methods and all data that were generated and analyzed in this study. Any raw data not explicitly shown are available from the corresponding author, subject to the approval of the institutional review board (IRB). Any identifiable patient information is protected according to the approved Soroka University Medical Center IRB protocol 0366-21-SOR. [Soroka University Medical Center IRB] [https://hospitals.clalit.co.il/soroka/he/ac-se/helsinki/Pages/default.aspx (accessed on 7 March 2025)] [0366-21-SOR].

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
