# Peer review of "Initial Implementation and Utilization of Cardiopulmonary Exercise Testing at a Pulmonary Department of an Academic Tertiary Care Center: An Overview"

_jcm, 2025, doi:10.3390/jcm14113676_

Round 1

Reviewer 1 Report

Comments and Suggestions for Authors

The manuscript entitled as “initial implementation and utilization of CPET at academic tertiary care center” emphasizing on understanding the pertinence of CPET in examining patients’ demographics, is impressive and holds merit. The study is not unique as there is a plethora of information already reported in literature from 2018-till date. Furthermore, lack of clinical outcome or preclinical examinations prevent the study from getting published in peer-reviewed journal.

  1. CPET cannot be considered as a gold standard for the risk assessment of any cardiovascular diseases especially- PAH, IPF, COPD, etc.
  2. The technique is helpful for diagnostic purposes, especially in people with unexplained dyspnea on exertion, because they will often show a normal workup at rest. Although CPET rarely pinpoints a specific diagnosis, it helps to narrow the differential diagnosis and guide further investigations. CPET results within normal limits can also limit further, unnecessary testing. In patients with known cardiac or pulmonary disease, identification of the cause(s) of exercise limitation can reveal potential therapeutic targets and guide therapy or rehabilitation strategies, especially when multiple comorbidities are present.
  3. The statistical data provided needs revaluation.
  4. Lack of preclinical and clinical investigations in the study limits the scientific rigor when it comes onto risk assessment potential of CPET.
  5. Furthermore, the data provided ensures the outcomes from 474 patients including male and females without elaborating inclusion & exclusion criterion for the patients selection without providing any comparative analysis in the distinction of primary indications for CPET in males over females or viceversa.

Author Response

  1. CPET cannot be considered as a gold standard for the risk assessment of any cardiovascular diseases especially- PAH, IPF, COPD, etc.

    Response:
    Thank you for this important clarification. We agree that CPET should not be broadly considered a gold standard for risk assessment across all cardiovascular or pulmonary conditions. We have revised the first paragraph of the Introduction to reflect this nuance, clarifying that CPET plays a supportive role in selected clinical contexts. Additionally, we have incorporated appropriate citations where CPET has been shown to contribute to risk stratification in specific populations

  2. The technique is helpful for diagnostic purposes, especially in people with unexplained dyspnea on exertion, because they will often show a normal workup at rest. Although CPET rarely pinpoints a specific diagnosis, it helps to narrow the differential diagnosis and guide further investigations. CPET results within normal limits can also limit further, unnecessary testing. In patients with known cardiac or pulmonary disease, identification of the cause(s) of exercise limitation can reveal potential therapeutic targets and guide therapy or rehabilitation strategies, especially when multiple comorbidities are present.

    Response:
     We have revised the first paragraph of the Introduction to clarify CPET’s role in narrowing the differential diagnosis, especially in cases of unexplained dyspnea, and its value in guiding further evaluation and therapy in patients with comorbid cardiopulmonary conditions.

  3. The statistical data provided needs revaluation.

    Response:
     All statistical analyses were re-evaluated, and we confirm that the models and results are accurate. If required, we can provide the results of the univariate analyses and a more detailed description of the model selection process beyond what is currently included in the Methods section.

  4. Lack of preclinical and clinical investigations in the study limits the scientific rigor when it comes onto risk assessment potential of CPET.

    Response:
    Thank you for this observation. This study was designed as a real-world, observational investigation focusing on CPET utilization patterns in a tertiary pulmonary department in the Middle East. While it does not include mechanistic or preclinical components, it incorporates clinical variables, pulmonary function test results, and CPET-derived physiological data—reflecting patient-level characteristics relevant to real-life implementation. We believe this pragmatic approach offers valuable insight into factors affecting CPET performance and potential barriers to achieving anaerobic threshold in routine practice.

    1. Furthermore, the data provided ensures the outcomes from 474 patients including male and females without elaborating inclusion & exclusion criterion for the patients selection without providing any comparative analysis in the distinction of primary indications for CPET in males over females or viceversa.

    Response:
    We have clarified the inclusion and exclusion criteria in the revised Methods section. Regarding sex-based analysis, univariate comparisons were conducted and showed no statistically significant association between sex and RER ≥ 1.1. During the multivariable model selection process, sex was excluded from the final model due to its limited predictive value and negative impact on model performance. In addition, as recommended, we have added Appendix B, which presents a detailed univariate comparison of baseline characteristics and referral indications by sex.

Reviewer 2 Report

Comments and Suggestions for Authors

This study is interesting, but it contains many weakness. 

Lack of a clear thesis and rationale – The introduction does not effectively present the main hypothesis of the study.  

Imprecise wording – Some parts of the text are too vague or ambiguous.  

Insufficient justification of methodology – The choice of research methods is not fully explained or justified.  

Incomplete references to the literature – Key studies in the field are missing from the citations.  

Lack of in-depth analysis of results – While data presentation is adequate, the interpretation is underdeveloped.  

Formatting inconsistencies – There are issues with the consistency of style, tables, and figures.  

Weak academic language – The text contains repetitions, overly colloquial expressions, or awkward sentence constructions.  

No discussion of study limitations – A section outlining potential limitations of the findings is missing.  

Weak conclusion – The summary fails to emphasize the most important findings and contributions of the study.  

Comments on the Quality of English Language

This study is interesting, but it contains many weakness. 

Lack of a clear thesis and rationale – The introduction does not effectively present the main hypothesis of the study.  

Imprecise wording – Some parts of the text are too vague or ambiguous.  

Insufficient justification of methodology – The choice of research methods is not fully explained or justified.  

Incomplete references to the literature – Key studies in the field are missing from the citations.  

Lack of in-depth analysis of results – While data presentation is adequate, the interpretation is underdeveloped.  

Formatting inconsistencies – There are issues with the consistency of style, tables, and figures.  

Weak academic language – The text contains repetitions, overly colloquial expressions, or awkward sentence constructions.  

No discussion of study limitations – A section outlining potential limitations of the findings is missing.  

Weak conclusion – The summary fails to emphasize the most important findings and contributions of the study.  

Conclusion  

This paper is of very low quality and does not meet the standards of scientific publishing. The numerous methodological, linguistic, and structural flaws indicate a lack of rigor in both research and writing. As a result, this manuscript is not suitable for publication in its current form.

Author Response

Lack of a clear thesis and rationale – The introduction does not effectively present the main hypothesis of the study

Response:
We have revised the final paragraph of the Introduction to clearly articulate the study’s rationale, objectives, and main hypothesis.

Imprecise wording – Some parts of the text are too vague or ambiguous.

Response:
 We thoroughly reviewed the manuscript and made substantial revisions to improve clarity, precision, and overall academic language throughout the text.

Insufficient justification of methodology – The choice of research methods is not fully explained or justified.  
Response:
Thank you for this important observation. We have added clarifying statements in the third paragraph of the Introduction and in the Discussion to better justify the selection of a retrospective, real-world observational design. The chosen methodology reflects the aim of evaluating CPET implementation in routine clinical practice and identifying real-life utilization patterns. Additionally, we have expanded on the Discussion to support the interpretation of our findings

Incomplete references to the literature – Key studies in the field are missing from the citations. 
Response:
Following an additional literature review, we have added several relevant citations to strengthen the background and support key statements in the Introduction and Discussion.

Lack of in-depth analysis of results – While data presentation is adequate, the interpretation is underdeveloped.  
Response:
Thank you for this helpful feedback. We have revised and expanded the Discussion section to provide a more in-depth interpretation of the results, including clinical context, possible explanatory mechanisms, and comparison with existing literature. These changes aim to enhance the scientific value and clarity of our findings.

Formatting inconsistencies – There are issues with the consistency of style, tables, and figures.
Response:
We have carefully reviewed the manuscript and made the necessary formatting corrections to ensure consistency across the text, tables, and figures.

Weak academic language – The text contains repetitions, overly colloquial expressions, or awkward sentence constructions.  
Response:
We have carefully revised the manuscript to improve academic tone, eliminate repetition, and correct awkward or informal phrasing throughout the text.

No discussion of study limitations – A section outlining potential limitations of the findings is missing. 

Response:
Thank you for your observation. A dedicated paragraph discussing the main limitations of the study has been added at the end of the Discussion section.

Weak conclusion – The summary fails to emphasize the most important findings and contributions of the study.

Response:
Thank you for this comment. We have revised the Conclusion section to better highlight the key findings and their clinical implications, as reflected in the revised abstract and final section of the manuscript. If there are additional points you believe should be emphasized, we would be grateful for further suggestions.

Reviewer 3 Report

Comments and Suggestions for Authors

Thank you for the opportunity to review the manuscript. CPET is a diagnostic tool that allows for a comprehensive assessment of the exercise capacity of patients, especially the functions of the circulatory and respiratory systems.

My suggestions for review:

  1. Please explain the abbreviations used in the abstract.
  2. Line 63-64 – Do patients really need to fast before the test? If we want to briefly describe the elements that make up the CTEP system, we should mention the respiratory gas analysis system.
  3. The data (after summing up) included in Figure 2 indicate 888 indications for CTEP insertion, from Figure 3 - 595 reasons for discontinuing the test. There were 434 patients studied, so where did this data come from?
  4. Figure 4. - how many tests were performed in particular years?
  5. The manuscript focuses on the assessment of predictors of achieving RER 1.1; however, CTEP provides much more data allowing to assess circulatory and pulmonary function. Have the authors considered the assessment of other indicators? Especially in the aspect of distinguishing between cardiogenic and pulmonary origin of dyspnea?

Author Response

  1. Please explain the abbreviations used in the abstract.
    Response:
    All abbreviations used in the abstract have been defined upon first mention, and a complete list of abbreviations has been added at the end of the abstract for clarity.

  2. Line 63-64 – Do patients really need to fast before the test? If we want to briefly describe the elements that make up the CTEP system, we should mention the respiratory gas analysis system.
    Response:
    We have revised the text to clarify that patients are advised to avoid heavy meals (not full fasting) prior to the test. Additionally, we have updated the description to explicitly include the respiratory gas analysis system as a core component of the CPET setup.

  3. The data (after summing up) included in Figure 2 indicate 888 indications for CTEP insertion, from Figure 3 - 595 reasons for discontinuing the test. There were 434 patients studied, so where did this data come from?

    Response:
    As noted in the figure captions, patients could have more than one referral indication or more than one reason for test termination. For example, a patient may be referred due to both dyspnea and ischemic heart disease, and test termination could involve both leg discomfort and shortness of breath. We have now emphasized this clarification more clearly in the main text and figure legends to avoid

  4. Figure 4. - how many tests were performed in particular years?
    Response:
    The number of tests performed each year is detailed in Appendix A. We have also added a clarifying statement in the Results section to ensure this information is clearly conveyed in the main text.

  5. The manuscript focuses on the assessment of predictors of achieving RER 1.1; however, CTEP provides much more data allowing to assess circulatory and pulmonary function. Have the authors considered the assessment of other indicators? Especially in the aspect of distinguishing between cardiogenic and pulmonary origin of dyspnea?

    Response:
    The secondary objectives of our study was to evaluate factors associated with test performance, including referral indications, pulmonary function parameters, and key CPET-derived variables. These were analyzed in relation to AT, RER ≥ 1.1, which was selected as a standardized and clinically accepted threshold to assess test effort and completion, as described in the Methods.

    We also conducted a similar analysis based on achieving ≥60% of predicted VOâ‚‚max, but ultimately focused the manuscript on a single primary effort-related parameter (RER) to ensure clarity and interpretability.

    We fully agree that CPET has great potential to differentiate between cardiogenic and pulmonary causes of dyspnea. Although this was beyond the scope of our current analysis, we plan to explore this diagnostic aspect in future work using additional indices such as VO2 MAX 

    Round 2

Reviewer 1 Report

Comments and Suggestions for Authors

All the comments have been well addressed by the authors. Therefore, I endorse the manuscript for publication.

Reviewer 2 Report

Comments and Suggestions for Authors

The authors improved the study, but still it does not reach the scientific standards for biomedical journals. 

Comments on the Quality of English Language

The authors improved the study, but still it does not reach the scientific standards for biomedical journals. 

Reviewer 3 Report

Comments and Suggestions for Authors

I believe that the manuscript can be published in its current form.